# Tumorigenicity of EGFR- and/or HER2-Positive Breast Cancers Is Mediated by Recruitment of Tumor-Associated Macrophages

**DOI:** 10.3390/ijms24021443

**Published:** 2023-01-11

**Authors:** Daeun You, Hyungjoo Kim, Yisun Jeong, Sun Young Yoon, Eunji Lo, Sangmin Kim, Jeong Eon Lee

**Affiliations:** 1Department of Breast Cancer Center, Samsung Medical Center, Seoul 06351, Republic of Korea; 2Department of Health Sciences and Technology, SAIHST, Sungkyunkwan University, Seoul 06351, Republic of Korea; 3Department of Surgery, Samsung Medical Center, Sungkyunkwan University School of Medicine, Seoul 06351, Republic of Korea

**Keywords:** EGFR, HER2, basal-like breast cancer, CCL2, tumor-associated macrophage

## Abstract

Basal-like breast cancer (BLBC) has a clinically aggressive nature. It is prevalent in young women and is known to often relapse rapidly. To date, the molecular mechanisms regarding the aggressiveness of BLBC have not been fully understood. In the present study, mechanisms of aggressiveness of BLBC involving EGFR and/or HER2 expression and interactions between tumor and tumor-associated macrophages (TAMs) were explored. The prognosis of breast cancer patients who underwent surgery at Samsung Medical Center was analyzed. It was found that the co-expression of EGFR and HER2 was associated with a worse prognosis. Therefore, we generated EGFR-positive BLBC cells with stable HER2 overexpression and analyzed the profile of secretory cytokines. Chemokine (C-C motif) ligand 2 (CCL2) expression was increased in HER2-overexpressed BLBC cells. Recombinant human CCL2 treatment augmented the motility of TAMs. In addition, the conditioned culture media of HER2-overexpressed BLBC cells increased the motility of TAMs. Furthermore, activation of TAMs by CCL2 or the conditioned culture media of HER2-overexpressed cells resulted in the production of pro-inflammatory cytokines, such as IL-8 and IL-1β. These observations reveal that CCL2 derived from EGFR and HER2 co-expressed BLBC cells can lead to increased TAM recruitment and the induction of IL-8 and IL-1β from recruited TAMs, triggering the tumorigenesis of breast cancer with the expression of both EGFR and HER2. Our findings demonstrate that EGFR+ and HER2+ BLBC aggressiveness is partially mediated through the interaction between BLBC and TAMs recruited by CCL2.

## 1. Introduction

Breast cancer is the most frequent cancer in women and is known to have a complex heterogeneity at the molecular level [1]. Basal-like breast cancer (BLBC) is a subtype of breast cancer characterized by the expression of genes that are expressed in the basal–myoepithelial cells of a normal breast [2]. Although the majority of BLBCs overlap with triple-negative breast cancer (TNBC) (defined as lacking the expression of the estrogen receptor (ER), progesterone receptor (PR), and human epidermal growth factor receptor 2 (HER2)), about 20% of BLBCs are known to express ER or HER2 [3]. Furthermore, previous studies have demonstrated that BLBCs with HER2 expression, which can be termed as a basal–HER2+ subtype, are associated with a worse prognosis compared to basal-like or HER2+ subtype breast cancer [4,5].

Among receptors of the ERBB family, EGFR is known as the preferred binding partner of HER2, following HER3 [6]. EGFR is associated with a poor prognosis in breast cancer. It is known to be overexpressed in BLBC containing TNBC [7]. When receptors of the ERBB family are activated by various ligands, they undergo receptor dimerization and kinase domain activation, leading to downstream signaling [8]. Previous reports have shown that breast cancer patients with the expression of both EGFR and HER2 have a poorer prognosis than other groups using 825 tumor samples [9] and frozen tumor sections from 670 primary breast cancer patients [10]. However, it is currently unclear why breast cancer patients with the expression of both EGFR and HER2 have a poorer prognosis than those with the expression of EGFR or HER2 alone.

Macrophages are innate immune cells known to be involved in host defense and tissue homeostasis maintenance. In cancer, tumor-associated macrophages (TAMs) are associated with tumor progression, tumor growth, and resistance to therapy [11]. TAMs consisting of over half of cells in the breast cancer microenvironment are sustained by both tissue-resident macrophages and circulating monocytes [12]. Monocytes recruited by colony stimulating factor 1 (CSF1) and chemokine (C-C motif) ligand 2 (CCL2) secreted by breast cancer cells can develop into TAMs under various tumor microenvironmental signals [13]. After monocytes are recruited and differentiated into M0 macrophages in the tumor microenvironment (TME), they are polarized towards M1-like macrophages which express antitumor characteristics or M2-like macrophages with pro-tumoral functions [14]. The majority of TAMs in the TME are M2-like macrophages activated by cytokines, such as interleukin (IL)-4, IL-10, and IL-13, and secrete cytokines known to support tumor progression [14,15].

The objective of this study was to determine the reason for the poor prognosis of breast cancer patients with the expression of both EGFR and HER2. Whether the tumor microenvironment including TAMs could play an important role in EGFR- and HER2-mediated cancer progression was also investigated. By analyzing the clinical data of surgical breast cancer patients, it was found that the simultaneous expression of EGFR and HER2 conferred a poorer prognosis compared to EGFR expression alone. Using an in vitro generated HER2-overexpressed EGFR+ cell line model, this study found that several cytokines, such as CCL2, were increased by HER2 overexpression, leading to a pro-tumoral microenvironment through the CCL2-mediated trafficking of TAMs in breast cancer.

## 2. Results

### 2.1. Co-Expression of EGFR and HER2 Correlates with Poor Survival in Breast Cancer Patients

The patient data were retrieved from the breast cancer patient clinical database of the Breast Cancer Center at Samsung Medical Center, Republic of Korea. There were a total of 950 invasive breast cancer patients between 1 January 1995 and 31 December 2002. They were classified by subtype and other clinicopathological properties (Table 1). The patients with positive EGFR expression (*n* = 115) were selected. The survival rates of those who were positive for EGFR only (EGFR+/HER2-/ER-, *n* = 92) and those who were positive for both EGFR and HER2 (EGFR+/HER2+/ER-, *n* = 23) were compared. As shown in Figure 1A,B, patients with both EGFR-positive and HER2-positive breast cancer showed significantly poor disease-free survival (DFS, *p* = 0.0229) and overall survival (OS, *p* = 0.019) compared to patients with EGFR-positive breast cancer.

### 2.2. Co-Expression of EGFR and HER2 Promotes Invasion and Proliferation

Next, we investigated why the survival rate of breast cancer patients with the simultaneous expression of EGFR and HER2 was poorer than that of patients with EGFR expression alone. To that end, we established an EGFR+ cell line, MDA-MB231, with stable HER2 overexpression. First, we checked the EGFR and HER2 mRNA expression levels (Figure 2A) and protein levels (Figure 2B) and confirmed HER2-specific overexpression in HER2-overexpressed cells. The invasiveness of the HER2-overexpressed cells was significantly increased compared to that of the Vec cells (Figure 2C). We also observed that HER2 overexpression promoted cell cycle progression based on flow cytometry analyses (Figure 2D). These results imply that EGFR- and HER2-positive breast cancer are more prone to tumor growth and metastasis than EGFR-positive breast cancer, which is consistent with the patient data shown in Figure 1.

### 2.3. Cytokine Secretion Profile Is Altered by HER2 Overexpression

Next, we hypothesized that secretory proteins could act as intermediators to regulate the proliferation and invasiveness of EGFR- and HER2-positive breast cancer. When we performed a human cytokine array using the conditioned culture media of MDA-MB231 Vec and HER2-overexpressed cells, our results show that the CCL2, CCL5, IL-6, and IL-8 secretion levels were increased in the HER2-overexpressed cells (Figure 3A). We also observed that the CCL2, CCL5, IL-6, and IL-8 levels were increased at the mRNA level (Figure 3B–E). To check the generalizability of the HER2-mediated increase in the cytokines in other EGFR+ cell lines, we overexpressed HER2 in another EGFR+ cell line, Hs578T, and found that HER2 overexpression led to increased levels of CCL2 in Hs578T cells as well (Appendix A). Furthermore, treatment with the culture media of HER2-overexpressed cells tended to increase CCL2 mRNA levels, implying a role of secreted factors in regulating CCL2 expression in HER2-overexpressed cells (Appendix A). As CCL2, CCL5, IL-6, and IL-8 were upregulated in the HER2-overexpressed cells, we tried to determine whether the expression levels of those cytokines were downregulated by HER2 inhibition using trastuzumab (TRZ), a HER2-neutralizing antibody. Unexpectedly, the TRZ treatment had no effect on the expression levels of CCL2, CCL5, IL-6, or IL-8 in the HER2-overexpressed cells (Appendix A).

### 2.4. CCL2 Expression Is Downregulated by Neratinib, a Pan-HER Inhibitor

From the previous results, we found that the TRZ treatment did not alter the expression levels of cytokines upregulated by HER2 overexpression. Therefore, we investigated the effect of neratinib, a pan-HER inhibitor, on CCL2 expression. Neratinib is a selective ERBB family inhibitor known to bind to and reduce the autophosphorylation of EGFR, HER2, and HER4 [16]. The transcript levels of CCL2, CCL5, IL-6, and IL-8 (Figure 4A, Appendix A) and protein levels of CCL2 (Figure 4B) induced by HER2 overexpression were dramatically decreased by neratinib treatment. In addition, the phosphorylation of EGFR and HER2 was completely suppressed by neratinib (Figure 4C). Furthermore, we performed HER2 knockdown to verify our previous findings. As shown in Figure 4D, HER2 knockdown decreased the endogenous CCL2 levels in MDA453 and BT474 cells. Based on these results, we suggest that pan-HER inhibitors, such as neratinib, could be more effective than TRZ in modulating CCL2 levels in breast cancer with EGFR and HER2 co-expression.

### 2.5. CCL2 Mediates Motility of TAMs

In previous studies, CCL2 was found to recruit various immune cells, including monocytes, immature dendritic cells, and natural killer cells [17]. A previous study has demonstrated that CCL2 produced by tumor and stromal cells in the TME can promote the recruitment of monocytes to the TME and the polarization of TAMs towards the pro-tumoral M2 type [18]. Among cytokines upregulated by HER2 overexpression, we focused on CCL2 because it was the most dramatically increased cytokine by HER2 overexpression. Considering previous reports and our results, we hypothesized that CCL2 could act as a mediator to determine the aggressiveness of EGFR+ HER2+ breast cancer.

To verify the functional role of CCL2 in the TME, we first generated M2-polarized macrophages using THP-1 human monocyte cells according to previously reported methods [19]. As shown in Figure 5A,B, the CD11b and CD163 expression was significantly increased in the M2-polarized macrophages. Furthermore, we tried to monitor the effects of CCL2 on the motility of M2-polarized macrophages. When CCL2 was used to treat M2-polarized macrophages, the migratory ability of the M2-polarized macrophages was dramatically increased (Figure 5C). Under the same experimental conditions, we analyzed the expression profile of secretory cytokines using the whole cell lysates of the M2-polarized macrophages with or without CCL2 treatment. We found that IL-8 and IL-1β levels were increased in the whole cell lysate of CCL2-treated M2-polarized macrophages (Figure 5D). Previous reports have shown that IL-8 and IL-1β, known to be proinflammatory cytokines, can promote cancer progression by supporting tumor proliferation, angiogenesis, and invasion [20,21]. These data indicate that CCL2 can increase the migration of M2-polarized macrophages and then accelerate the secretion of pro-tumoral cytokines, such as IL-8 and IL-1β, from M2-polarized macrophages. We also examined the effect of the conditioned culture media of Vec or HER2-overexpressed cells on M2-polarized macrophages. As expected, we observed that the migration ability of M2-polarized macrophages was increased by the conditioned culture media of HER2-overexpressed cells (Figure 5E). Although IL-1β expression was decreased, IL-8 production was also increased in the M2-polarized cells when treated with the conditioned culture media of HER2-overexpressed cells (Figure 5F). These data collectively suggest that CCL2, secreted from HER2-overexpressed cells, can increase the migratory abilities of M2-polarized macrophages and promote the secretion of pro-tumoral cytokines including IL-8.

Next, we investigated the recruitment of TAMs into the TME of the Vec or HER2-overexpressed cells. We injected Vec or HER2-overexpressed cancer cells into the second fat pads of NOD/SCID mice (Figure 6A). After 12 weeks, we found that TAM recruitment to the cancer site was significantly increased in mice injected with HER2-overexpressed cancer cells compared to that in mice injected with Vec cells through CD163 staining (Figure 6B).

## 3. Discussion

Among molecular subtypes of breast cancer, BLBC accounts for approximately 15% of all breast cancer cases and is known to be associated with poor patient survival. Approximately 65–72% of BLBC patients express EGFR [22,23]. In addition, HER2 amplification and overexpression are well-known predictive and prognostic biomarkers in breast cancer [24]. BLBC is known to mostly overlap with the triple-negative subtype with no expression of ER, PR, or HER2. Therefore, BLBC and HER2+ breast cancer could be commonly considered as mutually exclusive subtypes. However, about 20% of BLBC patients are known to express ER or HER2. In particular, BLBC with HER2 expression, the basal-HER2+ subtype, has been indicated in various reports. Laakso et al. (2006) have demonstrated that the basoluminal phenotype, which is distinguished by the partial expression of the basal cytokeratins CK5/14, is associated with HER2 amplification [25]. In addition, hormone receptor-negative breast cancers simultaneously expressed HER2 and basal markers, including CK5/6, CK14, and EGFR [4]. Torregrosa et al. (1997) have shown that breast cancer patients with EGFR and HER2 co-expression have the worst prognosis [9]. In addition, the co-amplification of EGFR and HER2 was inversely correlated with the DFS and OS of patients who have received hormonal therapy, radiotherapy, and chemotherapy [26]. In previous reports, the co-expression of EGFR and HER2 in breast cancer was associated with decreased patient survival and increased metastasis [10,27]. In accordance with previous reports, our results show that the simultaneous expression of EGFR and HER2 correlated with decreased survival using our breast cancer patient cohort database (Figure 1). However, the detailed mechanisms of poor prognosis related to the co-expression of EGFR and HER2 have not been well defined yet.

It is well known that various signaling pathways downstream of EGFR and HER2 are responsible for tumor progression in breast cancer. After undergoing homo- or hetero-dimerization, EGFR is known to activate Ras/Raf/mitogen-activated protein kinase, phosphatidylinositol-3-kinase/AKT/mammalian target of rapamycin, JNK, and phospholipase C [7]. The activation of these pathways can promote pro-tumoral functions, including proliferation, adhesion, survival, migration, invasion, and angiogenesis [7]. Heterodimerization among ERBB family proteins is known to result in amplified and diversified signaling compared to homodimerization [28]. The EGFR-HER2 heterodimer is known to be associated with increased metastasis [7]. EGFR-HER2 heterodimerization has been found to be increased in TRZ-resistant cells [29]. We have also reported that HER2-induced fibronectin expression is associated with TRZ resistance in breast cancer cells [30]. Consistent with these previous reports, our present results show that HER2 overexpression in EGFR+ breast cancer cells could lead to significantly increased cell invasion and cell cycle progression.

CCL2 is produced by various cells, including mesenchymal stem cells, leukocytes, fibroblasts, and tumor cells in the TME, acting as a chemoattractant for diverse types of immune cells [31]. CCL2 plays a role in promoting the recruitment of monocytes to pre-metastatic niches in breast cancer [32]. Metastasis-associated macrophages (MAMs), originating from monocytes that express CCR2, are known to contribute to the extravasation of tumor cells [32]. In addition, the inhibition of CCL2 using CCL2-specific antibodies downregulated monocyte recruitment and metastasis and increased survival in vivo [32]. CCL2 could induce the production of IL-1β in TAMs and lead to neutrophil expansion and systemic inflammation-mediated metastasis in breast cancer [33]. Here, we also analyzed secretory cytokines to delineate the underlying mechanism that induced cancer cell invasion and growth using the conditioned culture media of generated cells. Our results show that CCL2, CCL5, IL-6, and IL-8 were upregulated in the HER2-overexpressing cells. In particular, the levels of CCL2 expression induced by HER2 overexpression were significantly higher than those of other cytokines. An important aspect of our study was the focus on CCL2 and HER2-induced CCL2 expression, which accelerated the recruitment of TAMs. Collectively, our results demonstrate that the aggressiveness of breast cancer with EGFR and HER2 co-expression is associated with CCL2-induced recruitment of TAMs.

TAMs have received attention for being the most abundant tumor-infiltrating immune cells in the TME. Circulating monocytes can develop into non-polarized M0 macrophages by CSF1 [13]. Highly plastic M0 macrophages are differentiated into diverse subtypes of macrophages by various cytokines [13]. M1-like macrophages are induced by type 1 T helper cell cytokines, such as tumor necrosis factor (TNF), interferon-γ, and toll-like receptors. M2-like macrophages are stimulated by type 2 T helper cell cytokines, including IL-4, IL-10, and IL-13 [13]. TAMs are known to be more closely associated with tumor-promoting M2-like macrophages [15]. Generally, a high number of TAMs is associated with a poor prognosis in breast cancer [34,35]. Secreted cytokines derived from cancer cells can recruit circulating monocytes to the TME. Differentiated M2-like TAMs are known to secrete pro-tumoral cytokines, such as CCL18, TNF-α, and matrix metalloproteinases, that can promote cancer progression [15]. As described in previous reports [19], we also generated differentiated M2-like TAMs using THP1 monocytes. Recombinant human CCL2 or the conditioned culture media of HER2-overexpressed cells accelerated the motility of TAMs and increased the IL-8 and IL-1β production from activated TAMs (Figure 5). The secretion of IL-8 and IL-1β by TAMs could further enhance the aggressiveness of EGFR+ HER2+ breast cancer.

The tumor progression of TUBO cells (mouse breast cancer cells with HER2 overexpression) was correlated with M2-like macrophages. The depletion of TAMs significantly increased the anti-tumor efficacy of the anti-HER2 antibodies [36]. In addition, the knockdown of CCL2 and inhibition of its receptor resulted in delayed tumor growth in the MMTV-HER2 breast cancer model [37]. CCL2 could recruit CD206^hi^ macrophages and increase Wnt-1 production from those cells, which led to decreased levels of E-cadherin in the HER2+ cancer cells, resulting in the early dissemination of cancer cells [38]. These previous studies and our findings together imply an association between HER2 signaling and the CCL2-mediated recruitment of TAMs. In the present study, we found that HER2-induced CCL2 expression was decreased by neratinib but not by TRZ. Therefore, we propose that pan-HER inhibitors, such as neratinib and lapatinib, are effective drugs for treating breast cancer with EGFR and HER2 co-expression.

In contrast to the neratinib-mediated downregulation of CCL2, we could not observe any alterations in CCL2 levels when TRZ was used for treatment (Appendix A). The CCL5, IL-6, and IL-8 cytokines upregulated in the HER2-overexpressed cells (although to a lesser extent compared to CCL2) were not affected by TRZ either. Such a discrepancy might be due to the different modes of action between TRZ and neratinib. TRZ is known to act by inducing the degradation of HER2 and ADCC [39]. However, TRZ is unable to block the heterodimerization of HER2 with EGFR or HER3, which is known as a potential mechanism of TRZ resistance [40]. Neratinib, on the other hand, can inhibit the autophosphorylation of receptor tyrosine kinases by binding to the ATP-binding site of the intracellular domains of EGFR, HER2, and HER4. Therefore, the HER2-mediated expression of CCL2 might be mainly regulated by signaling downstream of EGFR and HER2 heterodimerization rather than the homodimerization of either receptor. Previously, Linde et al. (2018) have demonstrated that CCL2 is regulated downstream of NF-κB in MMTV-HER2 mice [38]. However, the mechanism that induces CCL2 expression in breast cancer with EGFR and HER2 co-expression remains to be further explored.

## 4. Materials and Methods

### 4.1. Clinicopathological Characteristics of Breast Cancer Patients

In the clinical database of the Breast Cancer Center at Samsung Medical Center, a total of 950 invasive breast cancer patients showed distinct metastasis at diagnosis. Patients’ clinicopathological factors, immunohistochemistry, biologic factors, and treatment modalities (such as type of operation and use of chemotherapy, radiotherapy, or hormone therapy) were obtained.

Pathological tumor stage was assessed according to the American Joint Committee on Cancer 6th Staging System. Estrogen receptor and progesterone receptor staining data were extracted from pathology reports. Staining was scored using the Allred score (AS), a method that could semi-quantitate the proportion of positive cells (scored on a 0–5 scale) and staining intensity (scored on a 0–3 scale), with a maximum score of 8. An AS > 2 was considered positive.

Tissue microarray paraffin blocks were made for each case. Slides were incubated with monoclonal antibodies against cytokeratin 5/6 (CK5/6, DAKO, Santa Clara, CA, USA, M7237, 1:100), epidermal growth factor receptor (EGFR, DAKO, M7239, 1:30), and human epidermal growth factor receptor 2 (HER2, DAKO, A0485, 1:250) for 1 h at room temperature (RT). HER2 fluorescent in situ hybridization (FISH) assays were performed using a PathVysion HER2 DNA Probe Kit (Abbott Molecular, Inc., Des Plaines, IL, USA) according to the manufacturer’s instructions.

Immunostaining for EGFR was interpreted as positive when at least 10% of tumor cells showed moderate to strong membrane staining. HER2 positivity was defined as an intensity of 3+ using immunohistochemistry (IHC) or as a gene amplification ratio ≥ 2.0 using FISH when IHC intensity was 1+ or 2+. Samples with no staining at all or membrane staining in < 10% of tumor cells had a score of “0”. The study protocol was approved by the Institutional Review Board (IRB) of the Samsung Medical Center [41].

### 4.2. Cell Culture

MDA-MB231 cells transfected with empty or human epidermal growth factor receptor 2 (HER2)-overexpression vector (retroviral pBMN) were generous gifts from Dr. Incheol Shin (Hanyang University, Seoul, Republic of Korea). Briefly, the empty vector or HER2-overexpression vector-transfected cells were selected by sorting enhanced green fluorescence protein-positive cells using flow cytometry (Becton–Dickinson (BD), San Diego, CA, USA). Generated MDA-MB231 Vec and HER2 breast cancer cell lines were maintained in Dulbecco’s Modified Eagle Medium supplemented with 10% fetal bovine serum (FBS, Hyclone, Logan, UT, USA), 100 IU/mL penicillin, and 100 µg/mL streptomycin. THP-1 human monocyte cells were obtained from Korea Cell Line Bank (Seoul National University, Seoul, Republic of Korea). Human THP-1 monocyte cell line was cultured in RPMI1640 containing 10% FBS (Hyclone), 100 IU/mL penicillin, and 100 µg/mL streptomycin.

### 4.3. HER2 Knockdown Using siRNAs

For HER2 knockdown, HER2-specific siRNAs (F: 5′-CCUGUGCCCACUAUAAGGA-3′, R: 5′-UCCUUAUAGUGGGCACAGG-3′) were transfected using Effectene (Qiagen, Valencia, CA, USA). SiRNAs with scrambled sequences were used as a control.

### 4.4. Establishment of M2-Polarized THP-1 Macrophages

To generate M2-polarized THP-1 macrophages, cells were treated with 320 nM phorbol-12-myristate-13-acetate (PMA) for 24 h. After 6 h of PMA treatment, cells were treated with 20 ng/mL IL-4 and 20 ng/mL IL-13 for an additional 18 h for differentiation. All cell lines were shown to be free of mycoplasma. Short tandem repeat analysis showed no cross-contamination between cell lines.

### 4.5. Quantitative Reverse Transcription Polymerase Chain Reaction (RT-qPCR)

Total RNA was prepared using TRIzol (Thermo Fisher Scientific, Waltham, MA, USA). Total RNA was used to synthesize cDNA with a cDNA synthesis kit (Thermo Fisher Scientific) according to the manufacturer’s protocol. RT-qPCR was performed with SYBR Green master mix (Bioline Ltd., London, UK) using a QuantStudio 6 Flex Real-Time PCR System (Thermo Fisher Scientific). Target gene expression was normalized to beta-actin (ACTB), a housekeeping gene. Relative gene expression (fold) was calculated using the comparative CT method (2^−ΔΔCT^) [42]. Thermal cycling conditions were 50 °C for 2 min, 95 °C for 10 min, followed by 40 cycles of 95 °C for 15 s, 60 °C for 15 s, and 72 °C for 15 s. Primer sequences used in RT-qPCR are listed in Appendix A.

### 4.6. Western Blotting

For immunoblotting, samples were lysed on ice using lysis buffer and boiled for 5 min in Laemmli sample buffer. Samples were loaded into SDS-polyacrylamide gels under denatured conditions and then transferred to polyvinylidene fluoride membranes (Millipore, Billerica, MA, USA). Membranes were blocked with skim milk at RT for 1 h and incubated with the following primary antibodies at 4 °C overnight (O/N): β-actin (Abfrontier, Seoul, Republic of Korea, LF-PA0207, 1:2000), t-EGFR (Abcam, Waltham, MA, USA, ab52894, 1:10,000), *p*-EGFR (Abcam, ab40815, 1:5000), t-HER2 (Santa Cruz Biotechnology, CA, USA, sc33684, 1:5000), *p*-HER2 (CST, Danvers, MA, USA, 2241S, 1:1000), and CD11b (Abcam, ab52478, 1:1000). After washing, membranes were incubated with horseradish peroxidase (HRP)-conjugated secondary antibodies (CST) at RT for 1 h. Protein signals were developed using ECL^TM^ prime reagent (GW Healthcare, Bucks, UK).

### 4.7. Cell Invasion/Migration Assays

Boyden chambers with/without Matrigel-coated filters (8 µm pore size) from BD were used for cell invasion/migration assays. For invasion assays, breast cancer cells were resuspended in fresh media (5 × 10^4^ cells/well) and loaded into the upper chamber coated with Matrigel. For migration assays, M2 cells were resuspended in fresh media (1 × 10^4^ cells/well) containing 100 ng/mL chemokine (C-C motif) ligand 2 (CCL2) recombinant protein and placed into the upper chamber. Fresh medium containing 5% FBS was added to the lower chamber. Cells were incubated at 37 °C for 48 h. After incubation, cells remaining in the upper chamber were removed. Invaded/migrated cells located on the bottom surface of the filter were fixed with 100% methanol, washed with phosphate-buffered saline (PBS), stained with toluidine blue dye, and analyzed using a Scanscope XT apparatus (Aperio Technologies, Vista, CA, USA).

### 4.8. Cell Cycle Analysis

Cell cycle was analyzed using propidium iodide (PI)-stained breast cancer cells. Harvested breast cancer cells were fixed with 70% ethanol at RT for 20 min. After fixation, cells were incubated with 100 µg/mL DNase-free RNase A (Thermo Fisher Scientific) at 37 °C for 30 min. After washing with PBS, the pellet was resuspended with fluorescence-activated cell sorting (FACS) buffer containing 50 µg/mL PI (Sigma-Aldrich, St. Louis, MO, USA) and analyzed using a BD FACS Calibur flow cytometer (BD).

### 4.9. Human Cytokine Array

Cytokine secretion levels in cell culture supernatants were analyzed using a human cytokine array kit (R&D systems, Minneapolis, MN, USA) according to the manufacturer’s instructions. Cell culture supernatant was collected for 24 h. Briefly, after blocking the membrane, supernatants were mixed with a biotinylated detection antibody cocktail and then incubated with the membrane at 4 °C O/N. After washing with 1X washing buffer, membrane was incubated with streptavidin-HRP for 30 min. The array was exposed to X-ray film using enhanced chemiluminescence. Cytokine levels were quantified based on positive control spots of the same membrane.

### 4.10. Enzyme-Linked Immunosorbent Assay (ELISA)

Cells were seeded into 6-well plates. Culture media were changed to media containing 1% FBS for 24 h. The amount of CCL2 in the conditioned media was measured using a human MCP-1 ELISA kit (KomaBiotech, Seoul, Republic of Korea) and a microtiter plate reader to measure absorbance at 450 nm.

### 4.11. Immunofluorescence Staining

Immunofluorescence staining was conducted using cells cultured on slides. Cells were fixed with 4% paraformaldehyde for 15 min at RT, permeabilized with 0.1% Tween 20 for 2 min, blocked with 10% BSA for 30 min at RT, and incubated with CD11b (Abcam, ab52478, 1:250) O/N at 4 °C. On the following day, slides were incubated with Alexa Fluor 488-conjugated secondary antibody (Invitrogen, Waltham, MA, USA, 1:250) at RT for 1 h and then were stained with Vectashield H-1200/DAPI (Vector Laboratories, Burlingame, CA, USA). Slides were carefully washed with PBS between every step. Images were collected with an LSM780 confocal laser-scanning microscope (Carl Zeiss, Zena, Germany).

### 4.12. Immunohistochemistry (IHC)

IHC staining of paraffin-embedded sections was performed by Samsung Medical Center Animal Pathology Core Laboratory. Briefly, formalin-fixed paraffin-embedded xenograft tissue sections were deparaffinized in xylene, dehydrated in graded alcohol, and hydrated in water. Sections were determined through H&E staining and then incubated with CD163 antibody (Thermo Fisher Scientific, MA5-11458, 1:50) at 37 °C for 1 h. After washing, samples were developed using an UltraView Universal DAB detection kit according to the manufacturer’s instructions. Images were digitally acquired using a Scanscope XT apparatus (Aperio Technologies, Vista, CA, USA) at 200× magnification.

### 4.13. Xenograft Studies

Nonobese diabetic/severe combined immunodeficiency (NOD/SCID) mice were used to establish xenograft models to observe TAM infiltration. Female NOD/SCID mice (Charles River, Kanagawa, Japan) at 6 to 8 weeks old were used in experiments. A total 1.5 × 10^6^ cells in 120 µL Matrigel (BD Biosciences, Bedford, MA, USA) were directly injected into secondary mammary fat pads. Approximately 12 weeks later, empty vector and HER2-overexpressed MDA-MD231 tumor tissues were harvested from NOD/SCID mice and fixed in formalin, followed by embedding in paraffin. Mice were kept in a pathogen-free animal house in accordance with the Institutional Animal Care and Use Committee of Samsung Biomedical Research Institute.

### 4.14. Statistical Analysis

Graphic data were generated using Microsoft Excel 2016 and GraphPad Prism 8 software (GraphPad Software, La Jolla, CA, USA). Student’s *t*-test (unpaired, two-tailed) was used to compared two groups. One-way ANOVA was employed to compare more than two groups. All experiments were repeated at least three times independently. Results are presented as mean ± standard error of the mean (SEM). Differences were considered significant at *p* < 0.05.

Clinical data were analyzed by the expression level of biological factors, such as EGFR, HER2, and hormone receptors, using IHC in breast tumor samples obtained from Samsung Medical Center, Republic of Korea. Survival curves were generated using the Kaplan–Meier method. Hazard ratio was estimated using Cox regression for disease-free survival/overall survival through multivariate analysis. Statistical significance was defined as *p* < 0.05. All statistical analyses were performed using SPSS Statistic 19.0 (IBM, Armonk, NY, USA).

## 5. Conclusions

In this study, we tried to investigate the cause of the difference in survival rates between BLBC patients with a co-expression of EGFR and HER2 and those with EGFR expression alone. The CCL2 expression level was increased by HER2 overexpression. However, HER2-induced CCL2 expression was decreased by neratinib but not by TRZ. Increased CCL2 enhanced TAM recruitment. Recruited TAMs showed increased levels of pro-tumoral cytokines, including IL-8 and IL-1β. These findings can be a potential mechanism determining the tumorigenicity of EGFR+ and HER2+ BLBC patients. Therefore, we believe that the pharmacological effects of pan-HER inhibitors, such as neratinib, are better than those of TRZ for treating BLBC patients with a co-expression of EGFR and HER2.

## Figures and Tables

**Figure 1 ijms-24-01443-f001:**
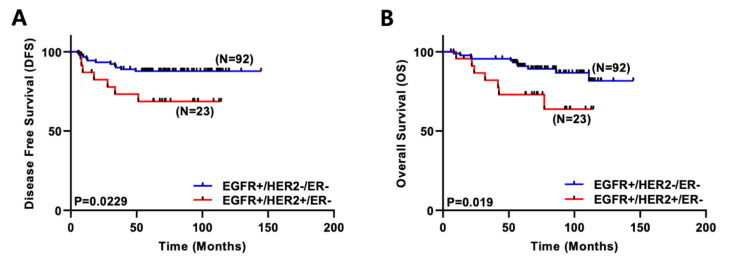
Co-expression of EGFR and HER2 correlates with poor survival in breast cancer patients. (**A**) Disease-free survival rates were compared between EGFR+/HER2-/ER- (*n* = 92) and EGFR+/HER2+/ER- (*n* = 23) patients (*p* = 0.0229). (**B**) Overall survival rates were compared between EGFR+/HER2-/ER- (*n* = 92) and EGFR+/HER2+/ER- (*n* = 23) patients (*p* = 0.019).

**Figure 2 ijms-24-01443-f002:**
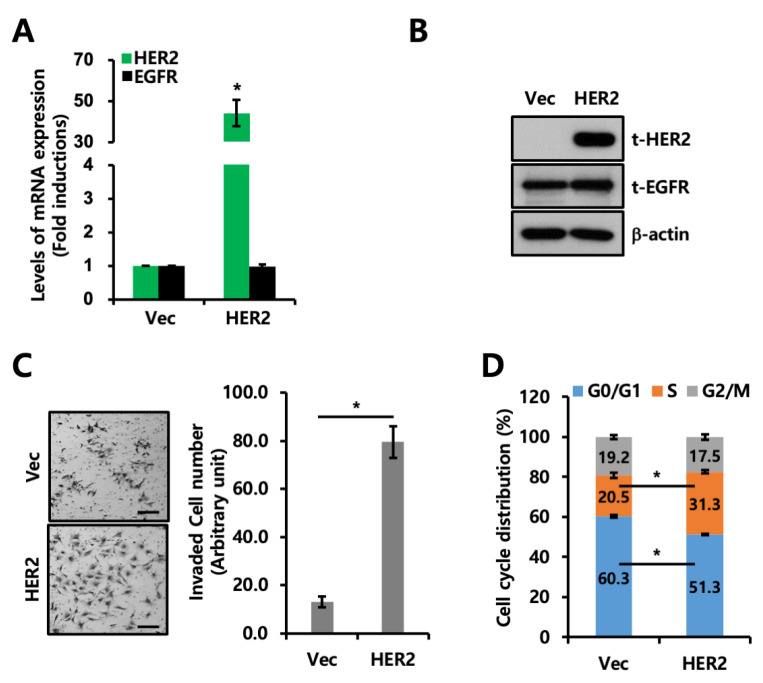
Co-expression of EGFR and HER2 promotes invasion and proliferation. (**A**) Transcript levels of *EGFR* and *HER2* in Vec and HER2-overexpressed MDA-MB231 cells were analyzed using quantitative reverse transcription polymerase chain reaction (RT-qPCR). Values were normalized to *ACTB*. (**B**) Levels of protein expression were analyzed using Western blotting with indicated antibodies. β-actin was used as a loading control. (**C**) Invasion assays were performed using MDA-MB231 Vec and HER2-overexpressed cells as described in the method section. The scale bar represents 200 μm. (**D**) Cell cycle analysis was performed using MDA-MB231 Vec and HER2-overexpressed cells. * *p* < 0.05.

**Figure 3 ijms-24-01443-f003:**
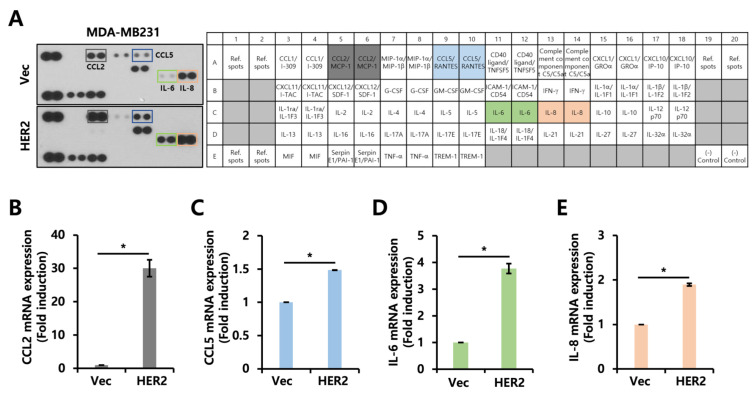
Cytokine secretion profile is altered by HER2 overexpression. (**A**) Human cytokine array was performed using conditioned culture media of MDA-MB231 Vec and HER2-overexpressed cells. (**B**–**E**) Transcript levels of *CCL2*, *CCL5*, *IL-6*, and *IL-8* in Vec and HER2-overexpressed MDA-MB231 cells were analyzed using RT-qPCR. Values were normalized to *ACTB*. * *p* < 0.05.

**Figure 4 ijms-24-01443-f004:**
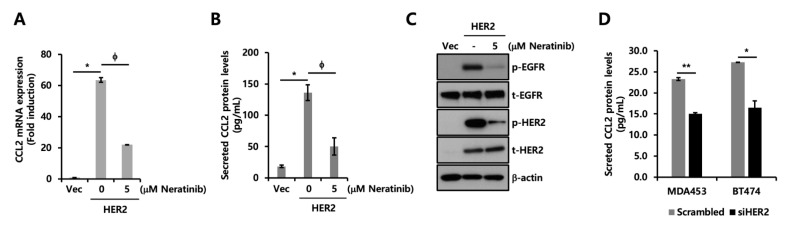
CCL2 expression is downregulated by neratinib. (**A**,**B**) Each cell line was treated with 5 μM neratinib for 24 h. (**A**) Transcript levels of *CCL2* were analyzed using RT-qPCR. Values were normalized to *ACTB*. (**B**) ELISA was performed using conditioned culture media of Vec and HER2-overexpressed MDA-MB231 cells. (**C**) Levels of protein expression were analyzed using Western blotting with indicated antibodies. β-actin was used as a loading control. (**D**) ELISA was performed using conditioned culture media of MDA453 and BT474 cells transfected with scrambled or HER2-specific siRNA. * *p* < 0.05; ** *p* < 0.01; ϕ *p* < 0.05.

**Figure 5 ijms-24-01443-f005:**
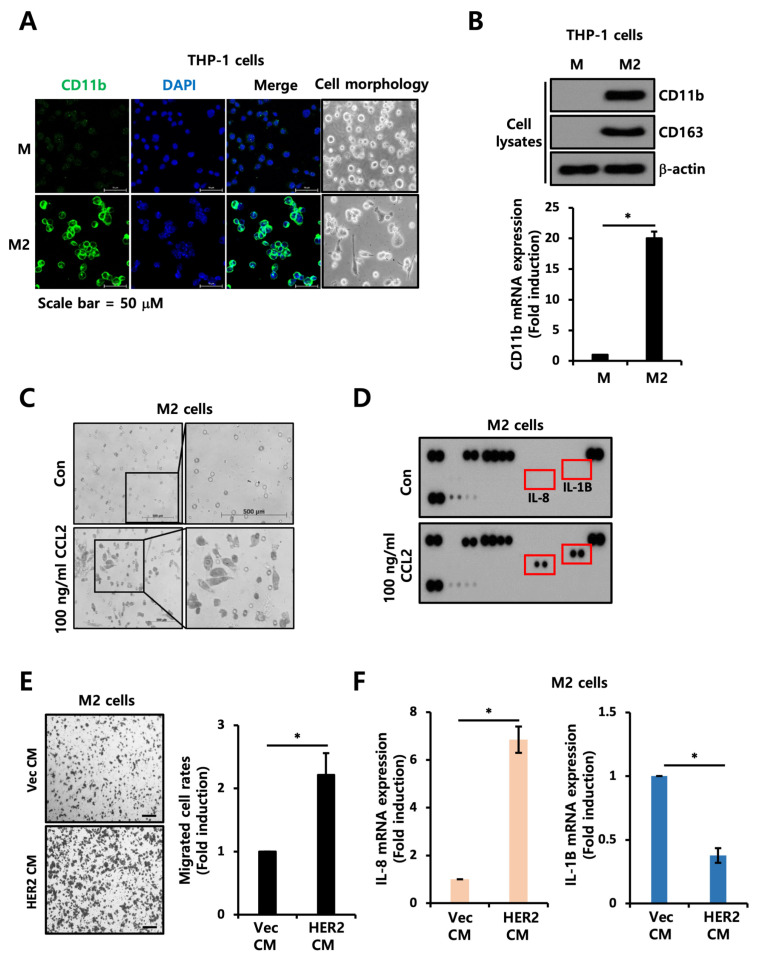
CCL2 mediates motility of TAMs. (**A**) Immunofluorescence confocal microscopy was performed using M0 and M2-differentiated THP-1 cells. Anti-CD11b primary antibody and Alexa Fluor 488-conjugated secondary antibody were used for staining. The nucleus was stained with DAPI. (**B**) Upper, levels of CD-11b and CD163 protein expression were analyzed using Western blotting with whole cell lysates from M0 and M2-differentiated THP-1 cells with indicated antibodies. β-actin was used as a loading control. Lower, transcript levels of *CD11b* in M0 and M2-differentiated THP-1 cells were analyzed using RT-qPCR. Values were normalized to ACTB. (**C**) Migration assays were performed as described in the method section. (**D**) Cytokine arrays were performed using whole cell lysates of control and CCL2-treated M2-differentiated THP-1 cells. (**E**) Migration assays were performed with M2-differentiated THP-1 cells treated with conditioned culture media from MDA-MB231 Vec and HER2-overexpressed cells. The scale bar represents 200 μm. (**F**) RT-qPCR showing transcript levels of *IL-8* and *IL-1B* in M2-differentiated THP-1 cells treated with conditioned culture media from MDA-MB231 Vec and HER2-overexpressed cells. Values were normalized to *ACTB*. * *p* < 0.05.

**Figure 6 ijms-24-01443-f006:**
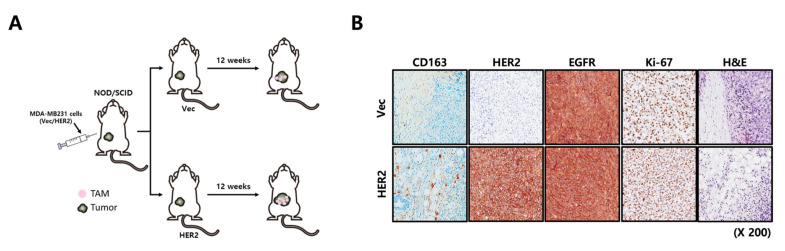
TAMs recruited into HER2-overexpressed tumors. (**A**) A schematic diagram illustrating the xenograft study in this figure: MDA-MB231 Vec or HER2-overexpressed cells were injected into the 2nd fat pad of NOD/SCID mice. Recruited TAMs were monitored after 12 weeks. (**B**) Recruited TAMs were analyzed using CD163 staining.

**Table 1 ijms-24-01443-t001:** Clinicopathological characteristics of breast cancer patients.

Variables	No. of Patients (%)
Age (years)	
<50	597 (62.8%)
≥50	363 (37.2%)
Menopause	
Pre.	588 (61.9%)
Post.	362 (38.1%)
ER	
Negative (−)	313 (35.9%)
Positive (+)	558 (64.1%)
PR	
Negative (−)	468 (53.1%)
Positive (+)	413 (46.9%)
EGFR	
Negative (−)	819 (86.3%)
Positive (+)	130 (13.7%)
HER2	
Negative (−)	715 (75.3%)
Positive (+)	235 (24.7%)
TP53	
Negative (−)	328 (41.8%)
Positive (+)	457 (58.2%)
Operation	
BCS ^1^	359 (37.8%)
MRM ^2^	591 (62.2%)
T stage	
T1	391 (41.2%)
T2	494 (52.0%)
T3	62 (6.5%)
T4	3 (0.3%)
*n* stage	
N0	498 (52.4%)
N1	248 (26.1%)
N2	116 (12.2%)
N3	88 (9.3%)
Stage	
I	258 (27.4%)
IIA	332 (35.2%)
IIB	135 (14.3%)
IIIA	130 (13.8%)
IIIB	2 (0.2%)
IIIC	85 (9.0%)
Nuclear grade	
Low	86 (9.1%)
Intermediate	471 (49.6%)
High	355 (37.4%)
Unknown	38 (4.0%)
CK 5/6	
Negative (−)	819 (86.2%)
Positive (+)	131 (13.8%)
Radiotherapy	
Yes	532 (61.6%)
No	332 (38.4%)
Unknown	15 (1.7%)
Chemotherapy	
Yes	794 (85.3%)
No	137 (14.7%)
Unknown	19 (2.0%)

*n* = 950, ^1^ BCS: breast-conserving surgery; ^2^ MRM: modified radical mastectomy.

## Data Availability

Not applicable.

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
