# Peer review of "Tumorigenicity of EGFR- and/or HER2-Positive Breast Cancers Is Mediated by Recruitment of Tumor-Associated Macrophages"

_ijms, 2023, doi:10.3390/ijms24021443_

Round 1
Author Response
Comments
Article no. Ms. Ref. No. IJMS (ISSN 1422-0067), Research Article entitled ‘Tumorigenicity of EGFR and/or HER2 positive breast cancers is mediated by recruitment of tumor associated macrophages’.
In the present study, authors have identified the role of HER2 overexpression in BLBC aggressiveness by modulating the recruitment of TAMs. These are a few novel findings and may also hold some translational relevance. However, I do notice many major important concerns that should be addressed to further enhance the manuscript quality:
Comment 1: It would be interesting if authors could analyze the PF and DFS of BLBC patients with different levels of M2-TAMs infiltration in tumor tissues. In addition, the authors need to properly include the Fig. 5F and Supplementary Fig. 2 labels in the text. Right now, Fig. 5F label is missing and the Supplementary Fig. 2 label is used to describe the Supplementary Fig. 1 content in the text.
→ We thank the reviewer’s careful reading. We also think that it would be interesting to analyze patient survival according to the levels of M2-TAM infiltration. However, we could not analyze the levels of M2-TAM infiltration in those tissues, since the tissue microarray (TMA) results were obtained from a previous study that was conducted several years ago and the tissues were not preserved entirely for further analysis. Although we tried to check the levels of M2-TAM infiltration in the remaining tissues but failed due to technical issues. We have properly modified the figure labels for Fig. 2 and the Supplementary Figures.
Comment 2: Authors need to provide some additional validation of their findings using an additional basal-like TNBC cell such as MDA-MB-468 by overexpressing HER2. How does author can explain these findings to counteract the findings reported by Oh et. al., 2015 (https://doi.org/10.1016/j.abb.2015.04.008) where they have shown that EGFR negates the proliferative effect of oncogenic HER2 in MDA-MB-231 cells? Authors should also knock down EGFR in these BLBC cells and then look for the downstream phenotypes
→ We appreciate the reviewer’s comment. We have included the results showing the effect of HER2 overexpression on an additional basal-like TNBC cell line, Hs578T and included the results as Supplementary Fig. 1. As expected, like MDA-MB231 cells, CCL2 expression is increased by HER2 overexpression in Hs578T cells.
Oh et al., (2015) demonstrated that HER2 overexpression in MDA-MB231 had no effect on cell growth using MTT assay. Analyzing cell growth through techniques such as MTT assays are poor in terms of accuracy. To accurately measure cell growth, we analyzed cell growth with the CellTrace Far Red Cell Proliferation Kit which utilizes in vitro labeling of cells to trace multiple generations. Our results showed that HER2 overexpression induces cell growth (data not shown). Although we were not able to perform EGFR knockdown experiments, Shen et al. have reported that basal level of CCL2 expression is significantly decreased by EGFR knockdown in gallbladder cancer cells which also decreased cell invasiveness (PMID: 31182136). Therefore, we believe that EGFR knockdown may decrease CCL2 expression and cell invasiveness in our experimental model as well.
Comment 3: Why does the author choose the NOD/SCID mice for this kind of study as it has been known that these mice contain defective macrophage (https://www.jax.org/jax-mice- and-services/find-and-order-jax-mice/nsg-portfolio). They should either use humanized mice or they can co-transplant or co-inject the HER2 overexpressing BLBC cells with THP1 differentiated M2-TAMs pre-incubated with the CM media of HER2 overexpressing BLBC cells. They should also analyze the tumor growth with lung or liver metastasis in vivo using HER2 overexpressing BLBC cells such as MDA-MB-231.
→ We thank the reviewer’s kind comment. We do acknowledge that the NOD/SCID mice has defective macrophages. However, to our knowledge, although previous reports (PMID: 8415751, 7995938, 12816997) show that the mice with NOD genetic background retains defective macrophages, data shows that it is a reduced functionality of the macrophages (reduced levels of CD86 and IL-1 secretion) rather than a loss of function. Therefore, we think that we could use the NOD/SCID model for showing the recruitment of macrophages to the tumor site, since the defects in macrophages of the NOD/SCID mice should not completely abrogate the homing of those cells. For the comment regarding the analysis of in vivo metastasis in HER2-overexpressed breast cancer, we also think it would be necessary to explore the effect of in vivo metastasis of HER2-overexpressed BLBCs. However, it is difficult to observe spontaneous metastasis in orthotopic MDA-MB231 xenograft models. To our knowledge, we have to use 4T1 syngeneic mouse models or MDA-MB231 LM2 cells, which are lung-metastasized MDA-MB231 cell lines; We do not have those resources and could not analyze in vivo metastasis of HER2-overexpressed EGFR+ breast cancer.
Comment 4: Authors need to perform the neutralization of either IL6 or IL8 to cross-validate their observations by performing in vitro or in vivo experiments.
→ Thank you for your good comment. Previous reports demonstrate that neutralization of IL-6 or IL-8 could prevent cancer growth and metastasis through inhibiting downstream signaling pathways that are involved in cancer-intrinsic and extrinsic modulation of tumor aggressiveness (PMID: 26840088, 29093275). We assume that neutralizing IL-6 or IL-8 in our models would show similar effects with those previous literatures. However, the reason we chose CCL2 among the chemokines that were upregulated by HER2-overexpression was because CCL2 was the chemokine that was altered the most in the HER2-overexpressed cells, and we focused on the role of CCL2 for the scope of our manuscript. Through performing additional experiments using PF4136309, a CCR2 antagonist, we found that CCR2 inhibition also leads to downregulation of in vitro cell growth and invasion of HER2-overexpressed cells (data not included in manuscript), further strengthening the role of CCL2 in HER2-overexpressed BLBCs. We also think it would be interesting to observe the effects of neutralizing of IL-6 or IL-8 in our model but could not perform the experiments since it seemed slightly beyond the scope of our manuscript.
Reviewer 2 Report
Strengths:
CCL2 derived from EGFR and HER2 co-expressed BLBC cells can lead to increased TAM recruitment and induction of IL-8 and IL-1β from recruited TAMs, triggering tumorigenesis of breast cancer with expression of both EGFR and HER2. EGFR+ and HER2+ BLBC aggressiveness is partially mediated through the interaction between BLBC and TAMs recruited by CCL2.
Weakness:
(a) In the Figure 1, patients with EGFR+/HER2-/ER- breast cancer showed significantly poor disease free survival and overall survival. Thus, the authors established an EGFR+ cell line, MDA-MB231, with HER2 overexpression. Also, is it a stable HER2 overexpressing cell? The following experiments are performed based on the established HER2 overexpressing cells. However, this lacks loss-of-function experiments as HER2 is knocked down in cells overexpressing EGFR and HER2.
(b) Neratinib is a selective ERBB family inhibitor known to bind to and reduce autophosphorylation of EGFR, HER2, and HER4. Expression levels of CCL2 mRNA (Fig. 4A) and protein (Fig. 4B) blocked by Neratinib treatment. Why the authors only examine the levels of CCL2 expression? Does neratinib have no effect on CCL5, IL-6, IL-8 and other transcripts after treatment?
(c) Why TRZ treatment did not alter expression levels of cytokines upregulated by HER2 overexpression. However, neratinib, a pan-HER inhibitor, action on CCL2 expression was observed. What’s difference and downstream action of the two drugs treatments?
(d) In the Figure 5, CD11b expression was significantly increased in M2-polarized macrophages. Actually, CD11b are total markers of macrophages. Is there more evidence on specific M2-polarized macrophages?
(e) Previous reports have shown that IL-8 and IL-1β known to be proinflammatory cytokines can promote cancer progression. It is questionnaire why the authors examined IL-8 and IL-1β mRNA expression in the conditioned medium? It seems unreasonable or no rationale to determine mRNA expression levels rather than secreted protein levels.
(f) In the Figure 6, the authors injected Vec or HER2-overexpressed cancer cells to 2nd-fat pads of NOD/SCID mice. However, animal models do not reflect the real immune system during cancer development. In addition, CD163 expression levels indicated M2 markers in IHC staining results. Is there infiltration of M1 macrophages in the tumor?
(g) EGFR-HER2 heterodimer is known to be associated with increased cancer progression. Is it sensitive to EGFR-TKI treatment in the downstream cytokine production and macrophage recruitment in the HER2-overexpressed MDA-MB231 cells?
(h) Is there any direct evidence that CCL2-mediated recruitment of M2 macrophages rather than other cytokines or growth factors derived from cancer cells?
Author Response
Comments and Suggestions for Authors
Strengths:
CCL2 derived from EGFR and HER2 co-expressed BLBC cells can lead to increased TAM recruitment and induction of IL-8 and IL-1β from recruited TAMs, triggering tumorigenesis of breast cancer with expression of both EGFR and HER2. EGFR+ and HER2+ BLBC aggressiveness is partially mediated through the interaction between BLBC and TAMs recruited by CCL2.
Weakness:
(a) In the Figure 1, patients with EGFR+/HER2-/ER- breast cancer showed significantly poor disease free survival and overall survival. Thus, the authors established an EGFR+ cell line, MDA-MB231, with HER2 overexpression. Also, is it a stable HER2 overexpressing cell? The following experiments are performed based on the established HER2 overexpressing cells. However, this lacks loss-of-function experiments as HER2 is knocked down in cells overexpressing EGFR and HER2.
→ Thank you for the excellent comment. We generated stable HER2-overexpressed cell line using a retroviral system. We totally agree with your suggestion to perform loss-of-function experiments of HER2 in EGFR+ HER2+ cells. Unfortunately, however, it was difficult to conduct loss-of-function experiments of HER2 due to our technical limitations. Therefore, we ask for your understanding.
(b) Neratinib is a selective ERBB family inhibitor known to bind to and reduce autophosphorylation of EGFR, HER2, and HER4. Expression levels of CCL2 mRNA (Fig. 4A) and protein (Fig. 4B) blocked by Neratinib treatment. Why the authors only examine the levels of CCL2 expression? Does neratinib have no effect on CCL5, IL-6, IL-8 and other transcripts after treatment?
→ We thank the reviewer's careful reading. Since CCL2 was the chemokine which was altered the most in the HER2-overexpressed cells, we focused on the effect of neratinib treatment on CCL2. We have also included the effect of neratinib treatment on CCL5, IL-6, and IL-8 levels as Supplementary Fig. 4 and mentioned the results in the manuscript as following: “Transcript levels of CCL2, CCL5, IL-6, and IL-8 (Fig. 4A, Supplementary Fig. 4) and protein levels of CCL2 (Fig. 4B) induced by HER2 overexpression were dramatically decreased by neratinib treatment.”
(c) Why TRZ treatment did not alter expression levels of cytokines upregulated by HER2 overexpression. However, neratinib, a pan-HER inhibitor, action on CCL2 expression was observed. What’s difference and downstream action of the two drugs treatments?
→ Thank you for your good comment. To elaborate the differential effects of TRZ and neratinib treatment in modulation of CCL2 expression levels, we have included the following sentences in the discussion section: “Such discrepancy might be due to different modes of action between TRZ and neratinib. TRZ is known to act by inducing degradation of HER2 and ADCC [39]. However, TRZ is unable to block heterodimerization of HER2 with EGFR or HER3, which is known as a potential mechanism of TRZ resistance [40]. Neratinib, on the other hand, can inhibit autophosphorylation of receptor tyrosine kinases by binding to the ATP-binding site of intracellular domains of EGFR, HER2, and HER4. Therefore, HER2-mediated expression of CCL2 might be mainly regulated by signaling downstream of EGFR and HER2 heterodimerization rather than homodimerization of either receptor.”
(d) In the Figure 5, CD11b expression was significantly increased in M2-polarized macrophages. Actually, CD11b are total markers of macrophages. Is there more evidence on specific M2-polarized macrophages?
→ We thank the reviewer's comment. We have also included the results of CD163 expression in Fig. 5 to show evidence for specific M2-polarization of the macrophages.
(e) Previous reports have shown that IL-8 and IL-1β known to be proinflammatory cytokines can promote cancer progression. It is questionnaire why the authors examined IL-8 and IL-1β mRNA expression in the conditioned medium? It seems unreasonable or no rationale to determine mRNA expression levels rather than secreted protein levels.
→ We appreciate the reviewer's comment. CCL2 secretion was significantly increased in the conditioned culture media of HER2-overexpressed cells compared to Vec control. Therefore, we examined the alteration of secretory cytokines by treating human recombinant CCL2 or the conditioned culture media of Vec or HER2-overexpressed cells to the M2-polarized macrophages. In particular, because CCL2 protein was increased in the conditioned culture media of HER2-overexpressed cells, it was difficult to separate the expression levels of secretory cytokines which originate from the M2-polarized macrophages and the treated conditioned culture media. Therefore, we observed the transcripts levels of IL-8 and IL-1β in the M2-polarized macrophages instead of measuring the secreted protein levels. Similar to recombinant CCL2 treatment, conditioned media of HER2-overexpressed cells also significantly increased IL-8 mRNA expression and cell motility of M2-polarized macrophages. Based on these results, we suggest that HER2-induced CCL2 expression triggers IL-8 secretion and cell motility of M2-polarized macrophages.
(f) In the Figure 6, the authors injected Vec or HER2-overexpressed cancer cells to 2nd-fat pads of NOD/SCID mice. However, animal models do not reflect the real immune system during cancer development. In addition, CD163 expression levels indicated M2 markers in IHC staining results. Is there infiltration of M1 macrophages in the tumor?
→ We thank the reviewer's kind comment. We also agree to the author’s comment that the immune system of NOD/SCID model may not reflect the real immune system during cancer development, since the NOD/SCID model has defective immune cells. However, to allow the growth of our cell line models, our options were restricted to using an immunocompromised in vivo model. Also, although the NOD/SCID model is known to have defective macrophages due to the NOD genetic background, we assume that the defectiveness (decreased levels of IL-10 secretion and CD86 expression) would not influence our objective to observe the HER2-mediated recruitment of macrophages. Previous studies have shown that MDA-MB-231 cells induce both M1- and M2-polarization of macrophages (PMID: 35960749). Therefore, we assume that there would also be M1 macrophages in the tumor microenvironment of our in vivo mouse model as well.
(g) EGFR-HER2 heterodimer is known to be associated with increased cancer progression. Is it sensitive to EGFR-TKI treatment in the downstream cytokine production and macrophage recruitment in the HER2-overexpressed MDA-MB231 cells?
→ We appreciate the reviewer's comment. EGFR-HER2 heterodimer is known to be associated with tumor aggressiveness and our cell line model has both expression of EGFR and HER2, which seemed to be responsible for the increased tumor aggressiveness-related traits such as enhanced growth, invasion, and CCL2 expression. Therefore, we have treated neratinib, which is a TKI for EGFR, HER2, and HER4, to the cells. We have observed that neratinib treatment leads to decreased production of CCL2, CCL5, IL-6, and IL-8, which are cytokines that were increased in the HER2-overexpressed cells.
(h) Is there any direct evidence that CCL2-mediated recruitment of M2 macrophages rather than other cytokines or growth factors derived from cancer cells?
→ We thank the reviewer's comment. Although not in the breast-cancer specific context, other cytokines rather than CCL2, among the cytokines that we found to be upregulated in HER2-overexpressed cells, were previously reported to influence recruitment of macrophages. For instance, CCL5 induced macrophage recruitment in a hepatic ischemia/reperfusion injury model (PMID: 28623253); IL-6 produced by NIH3T3 cells with Src overexpression increased M2 polarization of macrophages (PMID: 29707119). Therefore, other factors including CCL5 and IL-6 might have influence in recruitment of M2 macrophages. However, since CCL2 was the most dramatically induced cytokine by HER2-overexpression and previously well known for its monocyte-recruiting function in cancer, we assume that CCL2 is the main factor that induces recruitment of M2 macrophages in our model.
Round 2
Reviewer 1 Report
The authors must need to address all the comments by performing some additional important experiments, especially in vivo studies using HER2 overexpressing BLBCs. MDA-MB-231 are invasive cell lines and have been reported to cause metastasis to clinically relevant tissues such as lymph nodes, lungs, and bones.
Author Response
The authors must need to address all the comments by performing some additional important experiments, especially in vivo studies using HER2 overexpressing BLBCs. MDA-MB-231 are invasive cell lines and have been reported to cause metastasis to clinically relevant tissues such as lymph nodes, lungs, and bones.
→ We appreciate the reviewer's comment. We also think it would be informative to include results from additional in vivo experiments observing metastasis of HER2-overexpressing MDA-MB-231 tumors. However, to our knowledge, it is difficult to observe spontaneous metastasis from MDA-MB-231 xenografts (PMID: 23118918). Therefore, most of the studies observing metastasis of MDA-MB-231 cells use tail vein injection or metastatic derivative MDA-MB-231-LM2 cell lines (PMID: 16049480). Due to the limited time (10 days) given for our revision and the limited resources, we could not perform those experiments. We apologize for these limitations in our response, and we would appreciate your understanding in this matter.
Reviewer 2 Report
1. In this study, the authors mainly used only one breast cancer cell line, MDA-MB-231(EGFR+/HER2-/ER-, basal subtype). Actually, the oncogenic content may be varied in different cancer cells, resulting in different cancer progression. The authors overexpressed HER2 in another EGFR+ cell line, Hs578T, and found that 125 HER2-overexpression led to increased levels of CCL2 in Hs578T cells as well (Supplementary Fig. 1). Is it functionally identical to overexpression of HER2 in MDA-MB-231? Moreover, I think HER2 loss-of-function studies (e.g. si-RNA or shRNA knockdown) in EGFR+/HER2+/ER- breast cancer cell lines like SKBR-3, AU 565, BT-474 are not a very difficult experiment.
2. In this method, there is no mention of using a retroviral system to stably overexpress HER2 cell lines. Moreover, the subtitle of “Real time-polymerase chain reaction (PCR)” should be quantitative reverse transcription PCR.
3. In the revision, there is no staining result for CD163 expression in Fig. 5.
4. Notably, the MDA-MB-231 cell line is an epithelial, human breast cancer cell line. Thus, its tumor growth and microenvironment in NOD/SCID model cannot reflect real human or animal immune system.
Moreover, the authors referred that MDA-MB-231 cells induce both M1- and M2-polarization of macrophages (PMID: 35960749). Why the study revealed no M2 macrophages in vector group (Figure 6B)?
Author Response
- In this study, the authors mainly used only one breast cancer cell line, MDA-MB-231(EGFR+/HER2-/ER-, basal subtype). Actually, the oncogenic content may be varied in different cancer cells, resulting in different cancer progression. The authors overexpressed HER2 in another EGFR+ cell line, Hs578T, and found that 125 HER2-overexpression led to increased levels of CCL2 in Hs578T cells as well (Supplementary Fig. 1). Is it functionally identical to overexpression of HER2 in MDA-MB-231? Moreover, I think HER2 loss-of-function studies (e.g. si-RNA or shRNA knockdown) in EGFR+/HER2+/ER- breast cancer cell lines like SKBR-3, AU 565, BT-474 are not a very difficult experiment.
→ We appreciate the reviewer's comment. We have conducted experiments to knockdown HER2 in MDA453 and BT474 cells and found that knockdown of HER2 led to decreased levels of CCL2 in those cells. We have included those results as Figure 4D and mentioned the results in the manuscript as following: “Furthermore, we performed HER2 knockdown to verify our previous findings. As shown in Fig. 4D, HER2 knockdown decreased endogenous CCL2 levels in MDA453 and BT474 cells.”
- In this method, there is no mention of using a retroviral system to stably overexpress HER2 cell lines. Moreover, the subtitle of “Real time-polymerase chain reaction (PCR)” should be quantitative reverse transcription PCR.
→ We appreciate the reviewer's careful reading. In the methods section (4.2.), we have previously mentioned the retroviral system (pBMN) that was used to generate the MDA-MB231 HER2 cells. To add more information, we have added the following sentence: “Briefly, the empty vector or HER2-overexpression vector transfected cells were selected by sorting enhanced green fluorescence protein-positive cells by flow cytometry (BD).”. Also, we have changed “Real time-polymerase chain reaction (PCR) to “quantitative reverse transcription polymerase chain reaction (RT-qPCR)” throughout the manuscript.
- In the revision, there is no staining result for CD163 expression in Fig. 5.
→ We thank the reviewer's comment. We have previously added the CD163 staining results as Supplementary Figure 5. Now, we have moved the CD163 staining results to Figure 5B.
- Notably, the MDA-MB-231 cell line is an epithelial, human breast cancer cell line. Thus, its tumor growth and microenvironment in NOD/SCID model cannot reflect real human or animal immune system.
→ We appreciate the reviewer's comment. We also agree to the reviewer’s comment and think that it would be more appropriate to use a mouse model with a humanized immune system or engraft mouse breast cancer cell lines. However, due to the limited time (10 days) that was given for the revision, it was difficult for us to conduct in vivo experiments which required resources that are not currently available for us. Furthermore, since a study (PMID: 34824220) has used a similar experimental setting, which engrafted human breast cancer cell lines into mouse models, to show that human breast cancer cell-derived CCL2 recruited macrophages in SCID mouse, we carefully suggest that the in vivo results from our experimental setting is sufficient to support the main idea of the manuscript.
Moreover, the authors referred that MDA-MB-231 cells induce both M1- and M2-polarization of macrophages (PMID: 35960749). Why the study revealed no M2 macrophages in vector group (Figure 6B)?
→ We appreciate the reviewer's comment. In Figure 6B, there is also positive CD163 staining in the Vec tissue and staining intensity is increased in the HER2 tissue. Therefore, we have mentioned the results as following: “TAM recruitment to the cancer site was significantly increased in mice injected with HER2-overexpressed cancer cells compared to that in mice injected with Vec cells through CD163 staining (Fig. 6B).”
Round 3
Reviewer 1 Report
I do not have any concern.
Reviewer 2 Report
I don't have any other questions.